# Glycolysis and glutaminolysis cooperatively control T cell function by limiting metabolite supply to N-glycosylation

Lindsey Araujo[1], Phillip Khim[2], Haik Mkhikian[3], Christie-Lynn Mortales[1], Michael Demetriou[2]*

[1]Department of Microbiology and Molecular Genetics, University of California, Irvine, United States; [2]Department of Neurology and Institute for Immunology, University of California, Irvine, United States; [3]Department of Pathology and Laboratory Medicine, University of California, Irvine, United States

**Abstract** Rapidly proliferating cells switch from oxidative phosphorylation to aerobic glycolysis plus glutaminolysis, markedly increasing glucose and glutamine catabolism. Although Otto Warburg first described aerobic glycolysis in cancer cells >90 years ago, the primary purpose of this metabolic switch remains controversial. The hexosamine biosynthetic pathway requires glucose and glutamine for de novo synthesis of UDP-GlcNAc, a sugar-nucleotide that inhibits receptor endocytosis and signaling by promoting N-acetylglucosamine branching of Asn (N)-linked glycans. Here, we report that aerobic glycolysis and glutaminolysis co-operatively reduce UDP-GlcNAc biosynthesis and N-glycan branching in mouse T cell blasts by starving the hexosamine pathway of glucose and glutamine. This drives growth and pro-inflammatory $T_H17$ over anti-inflammatory-induced T regulatory (iTreg) differentiation, the latter by promoting endocytic loss of IL-2 receptor-$\alpha$ (CD25). Thus, a primary function of aerobic glycolysis and glutaminolysis is to co-operatively limit metabolite supply to N-glycan biosynthesis, an activity with widespread implications for autoimmunity and cancer.

*For correspondence: mdemetri@uci.edu

**Competing interests:** The authors declare that no competing interests exist.

## Introduction

Under anaerobic conditions oxidative phosphorylation is blocked, forcing cells to generate ATP by converting glucose to lactate via glycolysis. This is a highly inefficient use of glucose, generating only 2 ATP per glucose instead of 36–38 ATP per glucose with complete oxidation. Yet under aerobic conditions, rapidly dividing cells such as T cells paradoxically switch their metabolism from oxidative phosphorylation to aerobic glycolysis and glutaminolysis, where glucose is fermented to lactate despite the presence of oxygen (*Wang et al., 1976*; *Warburg et al., 1927*; *Warburg et al., 1924*) and primary carbon flux into the tricarboxylic acid cycle (TCA cycle) is switched from glucose to glutamine (*DeBerardinis et al., 2007*). The function of this metabolic switch has remained obscure since Otto Warburg first reported that cancer cells ferment glucose to lactate despite the presence of oxygen (aerobic glycolysis) over 90 years ago (*Warburg et al., 1927*; *Warburg et al., 1924*). Glucose metabolites generated by glycolysis are precursors for nucleotide, amino acid and lipid biosynthetic pathways, leading some to suggest that increased glucose uptake may serve to promote the biomass required for rapid cell division (*Lunt and Vander Heiden, 2011*; *Vander Heiden et al., 2009*). However, these biosynthetic pathways account for <10% of the increase in glucose uptake induced by the Warburg effect (*DeBerardinis et al., 2007*; *Hume et al., 1978*), implying other critical

functions. Alternatively, constraining oxidative phosphorylation in rapidly dividing cells may serve to limit cellular damage from reactive oxygen species (*Le et al., 2010*). This hypothesis is questioned by the observation that genetically blocking lactate production in lung cancer does not activate oxidative phosphorylation in vivo yet still induces disease regression (*Xie et al., 2014*).

In contrast to cancer, T cells provide an excellent physiologic model to investigate the normal function of aerobic glycolysis and glutaminolysis. Indeed, aerobic glycolysis and glutaminolysis have recently been suggested to promote pro-inflammatory IFNγ synthesis and $T_H1$ differentiation (*Chang et al., 2013*; *Gubser et al., 2013*; *Klysz et al., 2015*), respectively. Aerobic glycolysis also promotes anti-tumor T cell activity and differentiation into pro-inflammatory T helper 17 ($T_H17$) over anti-inflammatory T regulatory (Treg) cells (*Ho et al., 2015*; *Michalek et al., 2011*; *Shi et al., 2011*). Several of these studies implicated specific pathway enzymes or metabolites, such as GAPDH, α-ketoglutarate and phosphoenolpyruvate, for control of IFNγ translation, $T_H1$ differentiation and intracellular $Ca^{2+}$ flux, respectively. However, these mechanisms are not easily generalizable to other cell types and/or do not consider the global consequences of metabolite shifts or the combined effects of aerobic glycolysis and glutaminolysis.

Asn (N)-linked protein glycosylation utilizes both glucose (*Sharma et al., 2014*) and glutamine, yet surprisingly, this pathway has not been investigated as a target of aerobic glycolysis and/or glutaminolysis. Branched N-glycans bind galectins at the cell surface to simultaneously inhibit the endocytosis and signaling of multiple surface receptors and transporters, thereby controlling cell growth, differentiation and disease states such as autoimmunity, cancer and type 2 diabetes (*Demetriou et al., 2001*; *Dennis et al., 2009*; *Granovsky et al., 2000*; *Lau et al., 2007*; *Mkhikian et al., 2011*; *Ohtsubo et al., 2005*; *Partridge et al., 2004*; *Zhou et al., 2014*). In T cells, N-glycan branching regulates development, growth, differentiation and autoimmunity by altering T cell receptor clustering/signaling, surface retention/localization of CD4, CD8, CD45 and CTLA-4, and the differentiation into pro-inflammatory $T_H1$ over anti-inflammatory $T_H2$ cells (*Chen et al., 2009*, *2007*; *Demetriou et al., 2001*; *Lau et al., 2007*; *Morgan et al., 2004*; *Zhou et al., 2014*).

Branching in N-glycans depends on UDP-GlcNAc biosynthesis via the hexosamine pathway (*Dennis et al., 2009*; *Grigorian et al., 2011*, *2007*; *Lau et al., 2007*; *Mkhikian et al., 2016*). De novo synthesis of UDP-GlcNAc, the sugar-nucleotide donor substrate required by the N-glycan branching Golgi enzymes Mgat1, 2, 4 and 5, begins with the conversion of fructose-6-phosphate to glucosamine-6-phosphate by the rate-limiting enzyme glutamine-fructose-6-phosphate transaminase (GFPT, *Figure 1A*). Fructose-6-phosphate is derived from glucose by the action of hexokinase/glucokinase (HK) followed by glucose-6-phosphate isomerase (GPI) and is the critical entry step into glycolysis via the key regulatory enzyme phosphofructokinase 1 (PFK1) (*Figure 1A*). GFPT also requires glutamine as an amine donor to generate glucosamine-6-phosphate from fructose-6-phosphate. Thus, the hexosamine pathway may be in direct competition with glycolysis and glutaminolysis for fructose-6-phosphate and glutamine, respectively. Here, we test the hypothesis that aerobic glycolysis and glutaminolysis cooperatively starve the hexosamine pathway of both glucose and glutamine, thereby reducing de novo UDP-GlcNAc biosynthesis and N-glycan branching, the latter altering cell function.

## Results

### $T_H17$ cytokines lower UDP-GlcNAc biosynthesis and N-glycan branching

To initially explore our hypothesis, we investigated whether aerobic glycolysis regulates branching during the differentiation of T cells into pro-inflammatory $T_H17$ cells. Induction of $T_H17$ differentiation strongly induces aerobic glycolysis, but when glycolysis is inhibited, cell fate is switched from pro-inflammatory $T_H17$ to anti-inflammatory iTreg cells by an unknown mechanism (*Shi et al., 2011*). If aerobic glycolysis starves the hexosamine pathway of fructose-6-phosphate, $T_H17$ cytokines are expected to decrease branching via reduced biosynthesis of UDP-GlcNAc. To evaluate this possibility, we performed flow cytometry with L-PHA (*Phaseolus vulgaris, leukoagglutinin*), a plant lectin that binds specifically to $β1,6$ GlcNAc-branched N-glycans made by the Mgat5 enzyme and serves as a gauge of overall branching (*Cummings and Kornfeld, 1982*; *Demetriou et al., 2001*; *Zhou et al., 2014*). Strikingly, $T_H17$ inducing cytokines (TGFβ+IL-6+IL-23) acted synergistically to markedly reduce L-PHA binding in activated T cells by ~3–4 fold, despite each cytokine individually having

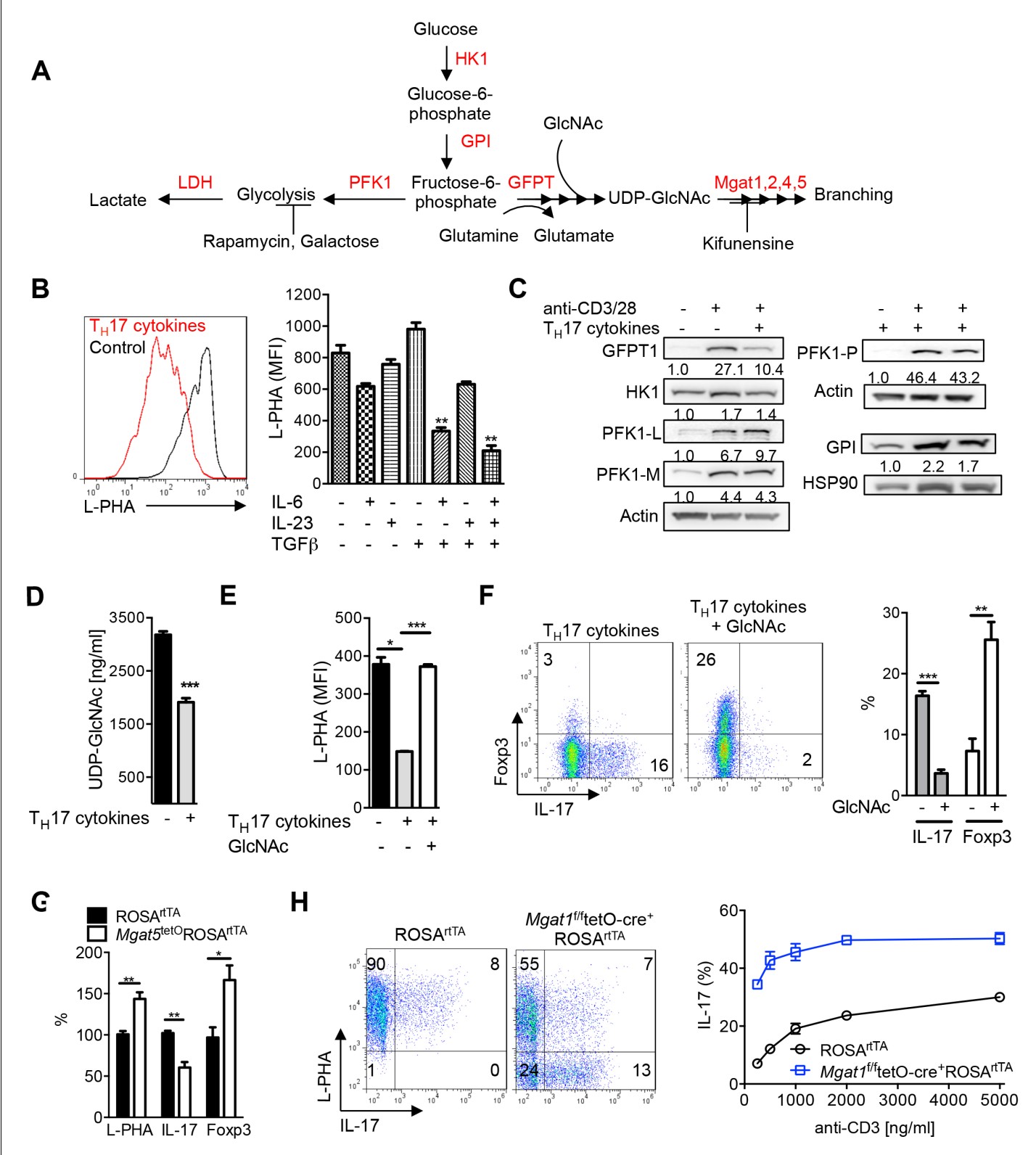

**Figure 1.** N-glycan branching controls T_H17 versus iTreg cell fate. (**A**) Fructose 6-phosphate and glutamine may be metabolized by glycolysis and glutaminolysis, respectively, or enter the hexosamine pathway to supply UDP-GlcNAc to the Golgi branching enzymes Mgat1, 2, 4 and 5. HK: hexokinase, GPI: glucose-6-phosphate isomerase, PFK1: phosphofructokinase1, LDH: Lactate dehydrogenase, GFPT: glutamine-fructose-6-phosphate transaminase. (**B–H**) Flow cytometry (**B,E–H**), Western blot (**C**) and LC-MS/MS (**D**) analysis of purified mouse splenic CD4+ T-cells activated with anti-

*Figure 1 continued on next page*

*Figure 1 continued*

CD3+anti-CD28 for 4 days (**B,E–H**) or 3 days (**C,D**) with T$_H$17 inducing conditions (TGFβ+IL-6+IL-23) or as indicated. PFK1-L (liver), PFK1-P (platelet), PFK1-M (muscle). (**G**) Co-incubation with doxycycline in vitro. (**H**) Doxycycline treatment in vivo, with *Mgat1*$^{f/f}$tetO-Cre$^+$ROSA$^{rtTA}$ cells in right panel gated on L-PHA$^−$ population. (**B,E–H**) gated on CD4$^+$. (**B,D–H**) Unpaired two tailed *t*-test with Welch's (**E**) and Bonferroni corrections (**B,E**). **p<0.01; ***p<0.001. Data are mean ± s.e.m of triplicate cultures and representative of n ≥ 3 experiments. MFI, mean fluorescence intensity.

The following figure supplements are available for figure 1:

**Figure supplement 1.** N-glycan branching controls T$_H$17 versus iTreg cell fate.

**Figure supplement 2.** N-glycan branching controls T$_H$17 versus iTreg cell fate.

minimal effects (*Figure 1B*, *Figure 1—figure supplement 1A*). Consistent with reduced branching T$_H$17 inducing cytokines also significantly reduced binding of E-PHA (*Phaseolus vulgaris, erthroagglutinin*) while increasing the binding of ConA (Concanavalin A), plant lectins that bind bisecting and high-mannose N-glycans, respectively (*Figure 1—figure supplement 1B*). Reduced branching should also decrease galectin binding and indeed, T$_H$17 inducing cytokines markedly reduced the binding of galectin-3 to activated T cells (*Figure 1—figure supplement 1B*). As only a ~20% change in branching is sufficient to alter T cell growth (*Lee et al., 2007*; *Mkhikian et al., 2011*; *Zhou et al., 2014*), the reduction induced by T$_H$17 inducing cytokines is highly significant. Although branching was reduced, mRNA expression of the Golgi branching genes *Mgat1* and *Mgat5* were unchanged or increased, consistent with reduced UDP-GlcNAc supply being primarily responsible for lowering branching (*Figure 1—figure supplement 1C*). Indeed, while T cell activation markedly increases protein expression of GFPT1 as well as the critical glycolytic enzymes HK1, GPI and PFK1 isoenzymes (liver, platelet and muscle), GFPT1 is uniquely and specifically down-regulated by T$_H$17 cytokines (*Figure 1C*). GFPT2 is an isoenzyme of GFPT1 but is not detectable by Western blot in T cells (data not shown). As GFPT1 and the three PFK1 isoenzymes all utilize fructose-6-phosphate, the reduction in GFPT1 induced by T$_H$17 cytokines should favor glucose flux into glycolysis over the hexosamine pathway. Indeed, UDP-GlcNAc production is reduced by T$_H$17 cytokines (*Figure 1D*, *Figure 1—figure supplement 1D*). Together, these data demonstrate that T$_H$17 cytokines reduce UDP-GlcNAc production, branching and GFPT1 expression, the rate-limiting enzyme for entry of fructose-6-phosphate into the hexosamine pathway.

## N-glycan branching induces a cell fate switch from T$_H$17 to iTreg

Next, we examined whether TGFβ+IL-6+IL-23 induced reductions in UDP-GlcNAc and branching was required for T$_H$17 differentiation. To test this hypothesis, we bypassed the effects of GFPT1 competition for fructose-6-phosphate by exploiting the hexosamine salvage pathway, where N-acetylglucosamine (GlcNAc) is used to generate UDP-GlcNAc directly (*Figure 1A*) (*Grigorian et al., 2007*; *Lau et al., 2007*). GlcNAc is metabolically inert within cells and does not enter glycolysis, the TCA cycle or the pentose phosphate pathway (*Wellen et al., 2010*). Supplementing T cells with GlcNAc reversed the reduction in branching induced by T$_H$17 cytokines and markedly inhibited T$_H$17 differentiation (*Figure 1E,F*). Remarkably, GlcNAc supplementation not only blocked T$_H$17 differentiation but also induced a cell fate switch to iTreg cells, despite the presence of T$_H$17-inducing cytokines (*Figure 1F*). The mannosidase I inhibitor kifunensine (*Figure 1A*) blocks branching (*Figure 1—figure supplement 1E*) and reversed the effects of GlcNAc supplementation, confirming that raising UDP-GlcNAc levels with GlcNAc supplementation blocked T$_H$17 and promoted iTreg differentiation by restoring branching (*Figure 1—figure supplement 1F*). Oral delivery of GlcNAc to mice with Experimental Autoimmune Encephalomyelitis, a model of multiple sclerosis, blocked disease progression, raised branching in T cells and suppressed T$_H$17 in vivo (*Grigorian et al., 2011*).

To confirm this result genetically, we utilized the tet-on system to generate a mouse with inducible expression of the Golgi branching enzyme Mgat5 (ROSA$^{rtTA}$*Mgat5*$^{tet}$). Indeed, directly raising branching via transgenic over-expression of *Mgat5* also induced a cell fate switch from T$_H$17 to iTreg cells despite T$_H$17-inducing cytokines (*Figure 1G*, *Figure 1—figure supplement 2A*). The magnitude of this change was less than that of GlcNAc supplementation, consistent with reduced de novo synthesis of UDP-GlcNAc by aerobic glycolysis primarily limiting branching. Directly inhibiting

branching should have the opposite effect of raising branching and indeed, blocking branching by culturing cells with kifunensine or by inducing deficiency of the branching enzymes Mgat1 (via doxy-cycline treatment of $Mgat1^{f/f}$tetO-Cre$^+$ROSA$^{rtTA}$ mice, *Figure 1—figure supplement 2B*) (*Zhou et al., 2014*) or Mgat5 (*Mgat5$^{-/-}$*) (*Demetriou et al., 2001*) all significantly enhanced T$_H$17 induction (*Figure 1H*, *Figure 1—figure supplements 1F* and *2C*). These data confirm that reduced UDP-GlcNAc production and branching induced by T$_H$17 cytokines is required for T$_H$17 differentia-tion and serves to block iTreg differentiation.

## N-glycan branching promotes IL-2Rα signaling to drive iTreg over T$_H$17 differentiation

Interleukin-2 (IL-2) suppresses T$_H$17 and promotes iTreg differentiation (*Davidson et al., 2007*; *Laurence et al., 2007*). As branching promotes surface retention of multiple receptors (*Dennis et al., 2009*), we investigated whether reduced branching limits IL-2 signaling via decreased surface retention of the IL-2 receptor. Blocking branching during T$_H$17 induction via *Mgat1* deletion markedly reduced surface expression and retention of CD25, the high-affinity alpha subunit of the IL-2 receptor (*Figure 2A*, *Figure 2—figure supplement 1A,B*). Up-regulation of branching via GlcNAc supplementation or *Mgat5* over-expression had the opposite effect, raising CD25 surface levels (*Figure 2B,C*, *Figure 2—figure supplement 1C,D*). In contrast, IL-2 cytokine levels were not

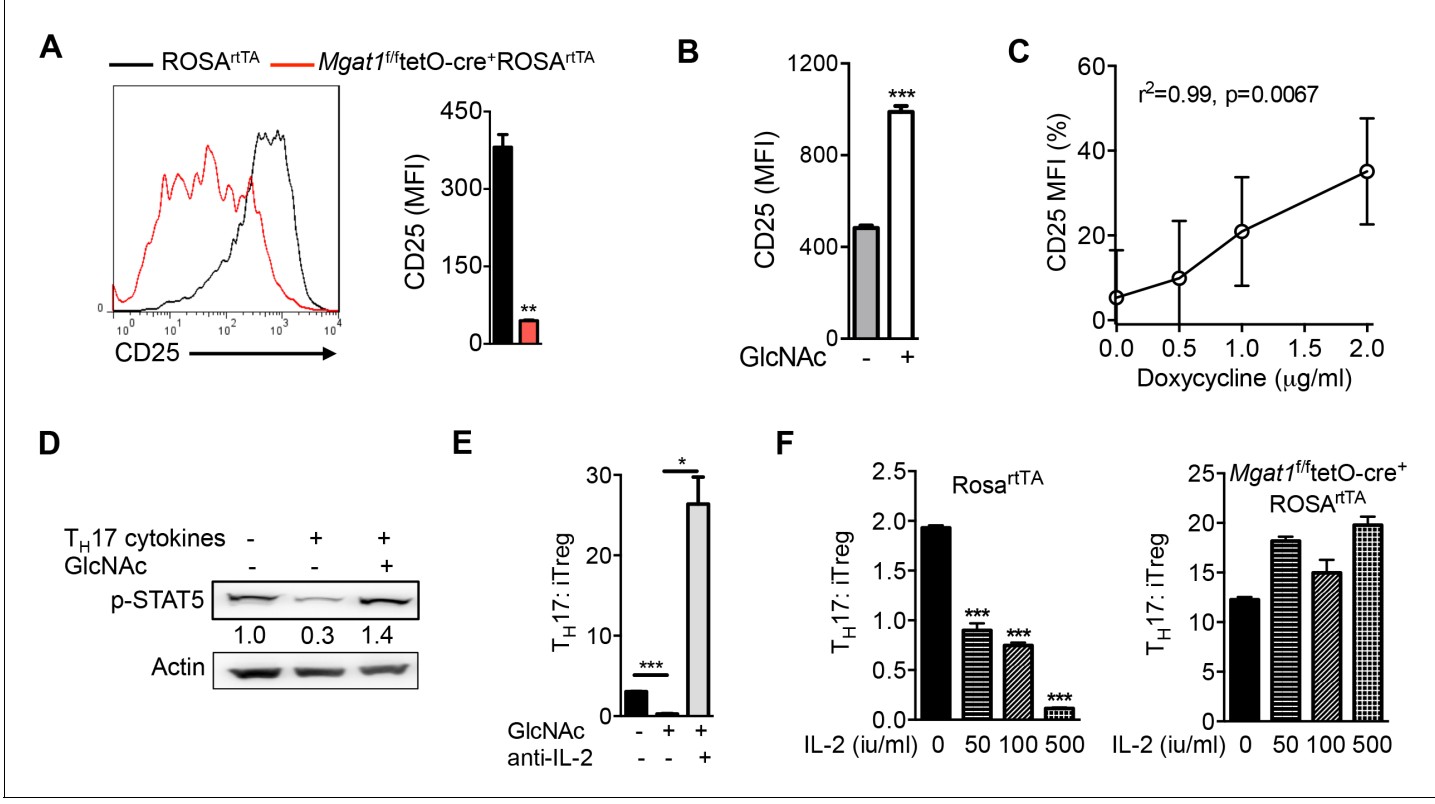

**Figure 2.** N-glycan branching controls T$_H$17 versus iTreg cell fate via IL-2Rα (CD25). (A–F) Flow cytometry (A–C,E,F) and Western blot (D) analysis of mouse splenic CD4$^+$ T-cells activated with anti-CD3+anti-CD28 under T$_H$17-inducing conditions (TGFβ+IL-6+IL-23) for 3 days (D) or 4 days (A–C,E,F); cells gated on CD4$^+$. (A,F) Doxycycline treatment in vivo with $Mgat1^{f/f}$tetO-Cre$^+$ROSA$^{rtTA}$ cells gated on L-PHA$^-$ population. (C) CD25 MFI in $Mgat5^{tetO}$ROSA$^{rtTA}$ CD4$^+$ T cells normalized to ROS$^{rtTA}$ CD4$^+$ T cells, both treated in vitro with doxycycline. (E,F) ratio of IL-17A$^+$ to FoxP3$^+$ CD4$^+$ cells. (A,B,E,F) Unpaired two-tailed *t*-test with Welch's (A,E) and Bonferroni corrections (E,F). (C) Linear regression. *p<0.05; **p<0.01; ***p<0.001. Data are mean ± s.e.m of triplicate cultures and representative of n ≥ 3 experiments. MFI, mean fluorescence intensity.

The following figure supplement is available for figure 2:

**Figure supplement 1.** N-glycan branching controls T$_H$17 versus iTreg cell fate via IL-2Rα (CD25).

significantly altered by GlcNAc or kifunensine (*Figure 2—figure supplement 1E*). The IL-2 receptor signals via STAT5 and this is markedly reduced by T$_H$17 cytokines (*Figure 2D*). GlcNAc supplementation restored pSTAT5 signaling despite T$_H$17 conditions (*Figure 2D*). Sequestering endogenous IL-2 with anti-IL-2 antibody blocked the ability of GlcNAc to switch cell fate from T$_H$17 to iTreg, suggesting that IL-2 is required for branching to promote iTreg over T$_H$17 differentiation (*Figure 2E*, *Figure 2—figure supplement 1F*). Moreover, inhibiting branching with *Mgat1* deficiency or kifunensine blocked the ability of exogenous IL-2 to induce a cell fate switch from T$_H$17 to iTreg (*Figure 2F*, *Figure 2—figure supplement 1G,H*). Taken together, these data indicate that T$_H$17 cytokine-induced down-regulation of GFPT1, UDP-GlcNAc and branching drive T$_H$17 over iTreg differentiation by lowering CD25 surface retention and IL-2 signaling.

## Aerobic glycolysis drives T$_H$17 over iTreg differentiation by lowering N-glycan branching

Next, we investigated whether aerobic glycolysis was required for T$_H$17 differentiation. Branching promotes surface retention of glucose transporters (*Lau et al., 2007*; *Ohtsubo et al., 2005*). Consistent with this, glucose uptake under T$_H$17-inducing conditions was enhanced by GlcNAc and inhibited by kifunensine, as measured by intracellular staining for the glucose analog 2-NBDG (*Figure 3—figure supplement 1A*). This suggests that GlcNAc supplementation does not directly inhibit aerobic glycolysis. Indeed, under T$_H$17 inducing conditions, GlcNAc supplementation did not significantly alter the glycolytic state of the cells prior to differentiation, as measured by the extracellular acidification rate (ECAR) and oxygen consumption rate (OCR) at day 2 of culture (*Figure 3A*, *Figure 3—figure supplement 1B*). This suggests that catabolism of fructose-6-phosphate to lactate is not necessary to drive T$_H$17 differentiation. To confirm this, we directly limited glucose flux into glycolysis by forcing oxidative phosphorylation using either the glycolytic inhibitor rapamycin or media supplemented with galactose instead of glucose (*Chang et al., 2013*). Rapamycin inhibits mTOR (mechanistic Target of Rapamycin) signaling via binding to mTOR complex 1, thereby reducing HIF-1α expression (*Land and Tee, 2007*), a transcription factor that up-regulates glycolysis enzymes including HK, PGM and PFK1 (*Semenza et al., 1994*). Galactose is much more slowly converted to pyruvate than glucose, forcing cells to up-regulate oxidative phosphorylation at the expense of glycolysis (*Chang et al., 2013*). Rapamycin and galactose both lowered aerobic glycolysis, enhanced branching and drove iTreg over T$_H$17 differentiation (*Figure 3B–G*). Importantly, the latter was reversed with *Mgat1* deficiency or kifunensine (*Figure 3F,G*, *Figure 3—figure supplement 1C,D*), demonstrating that the increase in branching induced by blocking glycolysis was necessary to switch cell fate from T$_H$17 to iTreg. Rapamycin had little effect on GFPT1 levels, while galactose marginally increased GFPT1, consistent with glycolysis largely acting by metabolically limiting flux of fructose-6-phosphate to the hexosamine/branching pathway rather than altering enzyme expression (*Figure 3E*). Importantly, kifunensine did not reverse the inhibition of aerobic glycolysis induced by rapamycin and galactose (*Figure 3B,C*), establishing that T$_H$17 differentiation does not require aerobic glycolysis beyond its effects on branching. 2-Deoxy-D-glucose is another commonly used inhibitor of glycolysis; however, this molecule is also 2-deoxy-D-mannose and inhibits N-glycan biosynthesis via incorporation of 2-deoxy-D-mannose into the lipid-linked oligosaccharide precursor (*Zhang et al., 2014*). Indeed, 2-deoxy-D-glucose/2-deoxy-D-mannose reduced branching and promoted T$_H$17 differentiation (*Figure 3—figure supplement 1F,G*), consistent with branching being dominant over aerobic glycolysis in controlling T$_H$17 differentiation. Taken together, these data demonstrate that a primary function of aerobic glycolysis during T$_H$17 differentiation is to lower branching by starving the hexosamine pathway of fructose-6-phosphate substrate availability.

## Oxidative phosphorylation drives iTreg differentiation by enhancing N-glycan branching

iTreg cells induced by TGFβ alone are non-glycolytic and utilize lipid oxidation for ATP production (*Michalek et al., 2011*; *Shi et al., 2011*). Based on our model that glycolysis and the hexosamine pathway compete for fructose-6-phosphate, lipid oxidation should favor flux of fructose-6-phosphate into the hexosamine pathway over glycolysis and thereby increase UDP-GlcNAc, branching and iTreg differentiation via IL-2/CD25/STAT5 signaling. Indeed, abolishing branching via *Mgat1* deficiency virtually eliminated iTreg differentiation induced by TGFβ (*Figure 4A*, *Figure 4—figure supplement*

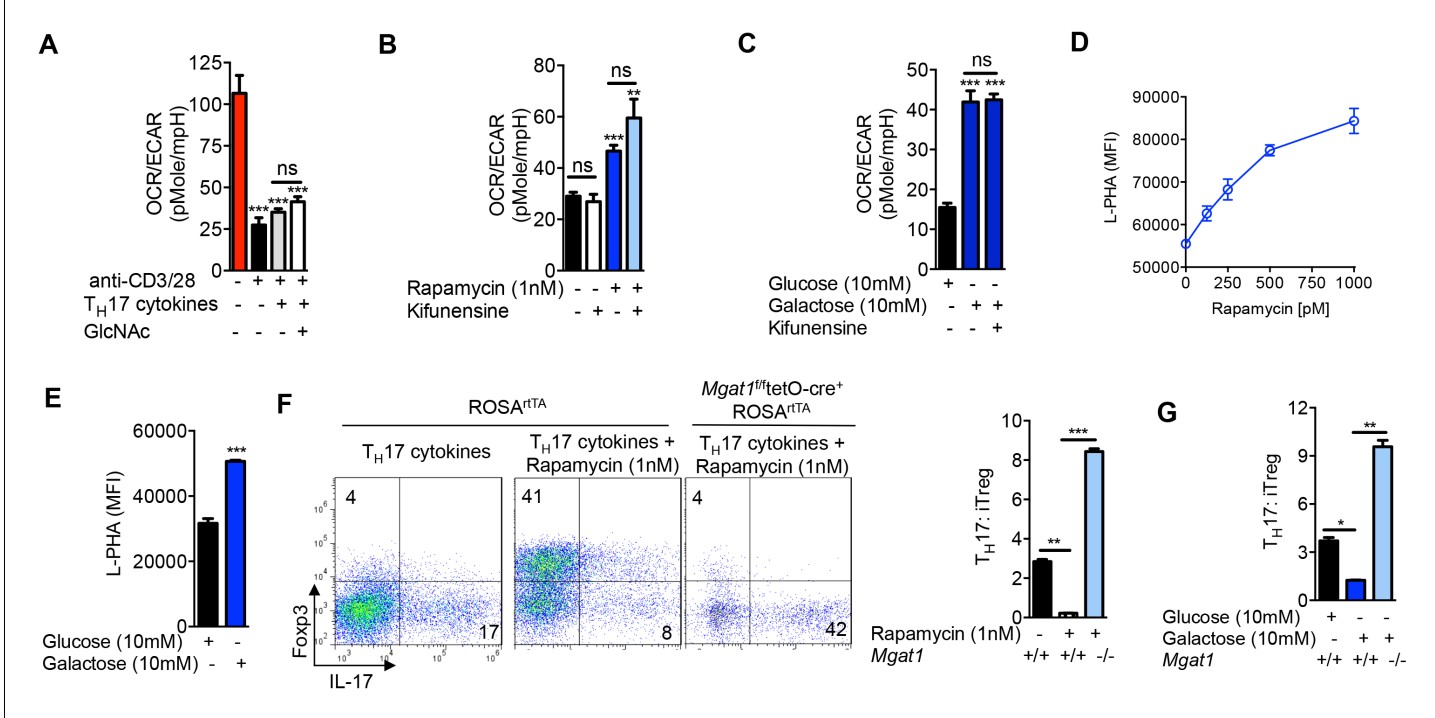

**Figure 3.** Glycolysis promotes $T_H17$ over iTreg cell fate by inhibiting N-glycan branching. (A–C) Ratio of the oxygen consumption rate (OCR) to the extracellular acidification rate (ECAR) of purified splenic CD4$^+$ T cells at rest or activated with anti-CD3+anti-CD28 for 2 days with $T_H17$ cytokines (TGF$\beta$ +IL-6+IL-23) or as indicated. (D–G) Flow cytometry analysis of purified mouse splenic CD4$^+$ T-cells activated with anti-CD3+anti-CD28 for 4 days under $T_H17$-inducing conditions (TGF$\beta$+IL-6+IL-23). (F (right panel), G) ratio of IL-17A$^+$ to Foxp3$^+$ CD4$^+$ T cells, with $Mgat1^{f/f}$tetO-Cre$^+$ROSA$^{rtTA}$ cells gated on L-PHA$^-$ population. *p<0.05; **p<0.01; ***p<0.001. NS, not significant. (A–G), Unpaired two-tailed $t$-test with Welch's (B,C,F,G) and Bonferroni correction (A–C,F,G). Data are mean ± s.e.m and n = 3. MFI, mean fluorescence intensity.

The following figure supplement is available for figure 3:

**Figure supplement 1.** Glycolysis promotes $T_H17$ over iTreg cell fate by inhibiting N-glycan branching.

1A). iTreg differentiation was also inhibited by *Mgat5* deficiency and kifunensine (*Figure 4B,C*), while raising branching with GlcNAc had the opposite effect (*Figure 4C*, *Figure 4—figure supplement 1B*). Kifunensine reversed the effects of GlcNAc, confirming GlcNAc promoted iTreg by raising branching (*Figure 4C*). Treatment with etomoxir (ETX), a specific inhibitor of fatty acid oxidation (*Michalek et al., 2011*), should up-regulate glycolysis and thereby limit flux of fructose-6-phosphate into the hexosamine pathway. Indeed, etomoxir decreased branching, CD25 surface expression and iTreg generation; phenotypes reversed by GlcNAc supplementation (*Figure 4D–F*). Taken together, these data confirm that oxidative phosphorylation promotes iTreg differentiation by limiting utilization of fructose-6-phosphate by glycolysis and thereby promoting flux of fructose-6–phosphate into the hexosamine pathway.

## Aerobic glycolysis promotes T cell growth by lowering N-glycan branching

Branching inhibits T cell growth by reducing T cell receptor clustering/signaling and promoting surface retention of the growth inhibitor CTLA-4 (*Demetriou et al., 2001*; *Lau et al., 2007*; *Mkhikian et al., 2011*). In the absence of polarizing cytokines, TCR stimulation also increases glucose uptake and induces aerobic glycolysis. However, in contrast to $T_H17$ cells, GFPT1 is markedly up-regulated in neutral T cell blasts (*Figure 1C*). Thus, increased GFPT1 should be better at competing with PFK1 for the increased supply of fructose-6-phosphate to promote UPD-GlcNAc production. Coupled with increased expression of multiple Golgi branching enzymes in T cell blasts (*Chen et al., 2009*), this should promote branching in neutral T cell blasts relative to resting cells.

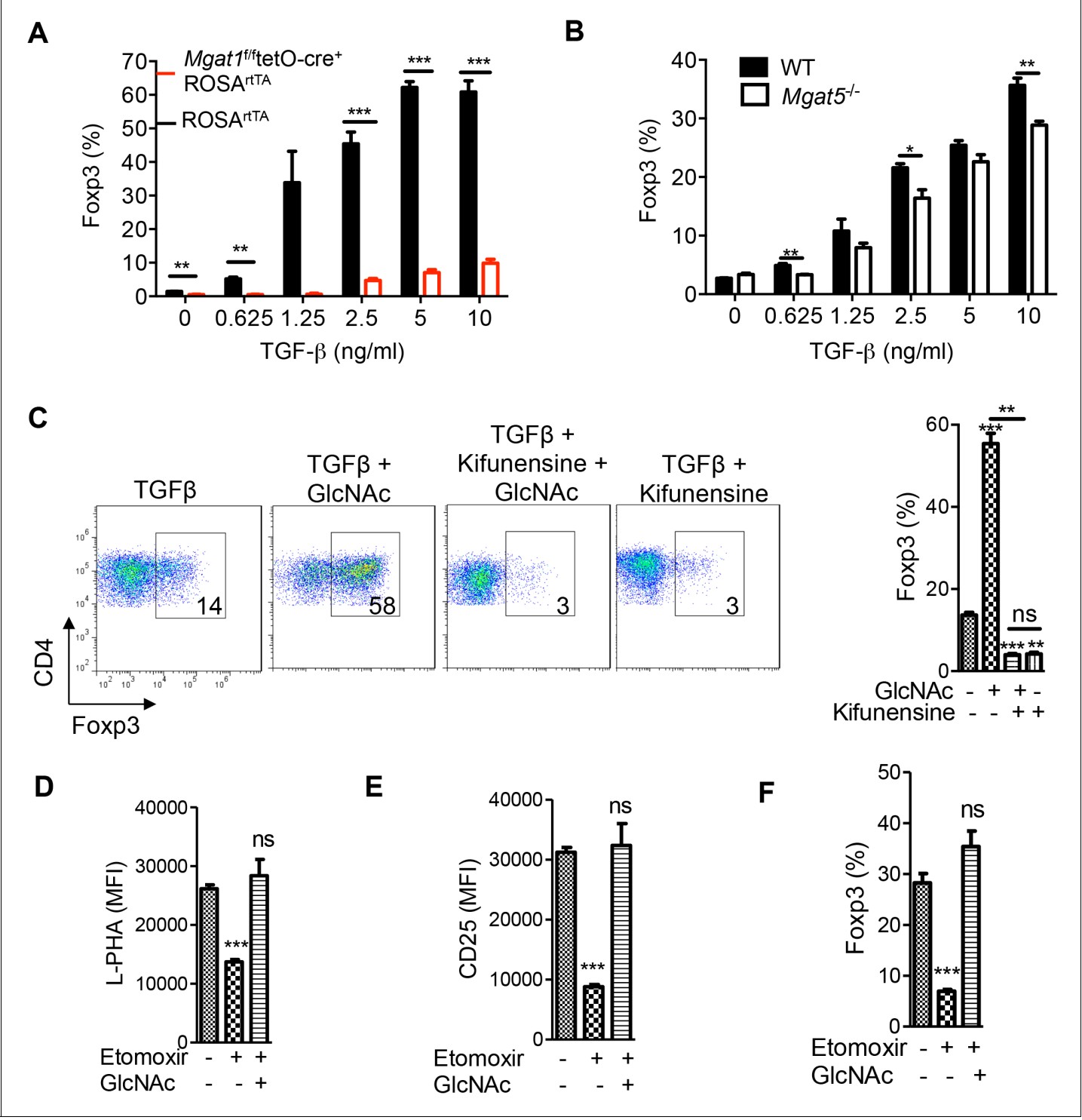

**Figure 4.** Oxidative phosphorylation promotes iTreg generation by enhancing N-glycan branching. (**A–F**) Flow cytometry of purified splenic CD4+ T cells activated with anti-CD3+anti-CD28 and TGFβ1 for 4 days; gated on CD4+. (**A**), *Mgat1f/f*tetO-Cre+ROSArtTA cells gated on L-PHA- population from mice treated with doxycycline in vivo. *p<0.05; **p<0.01; ***p<0.001. (**A–F**) Unpaired two-tailed *t*-test with Bonferroni (**C–F**) and Welch's (**C,E,F**) correction. Data are mean ± s.e.m and n = 3. MFI, mean fluorescence intensity.

The following figure supplement is available for figure 4:

**Figure supplement 1.** Oxidative phosphorylation promotes iTreg generation by enhancing N-glycan branching.

Indeed, we previously observed increased UDP-GlcNAc production and branching in non-polarized T cell blasts relative to resting naive T cells (*Lau et al., 2007*). However, inhibiting aerobic glycolysis under neutral conditions should further drive fructose-6-phosphate into the hexosamine pathway to promote branching and limit growth of T cell blasts. Indeed, rapamycin and galactose enhanced branching and inhibited T cell growth, effects reversed with the Golgi branching inhibitor kifunensine (*Figure 5A–C*, *Figure 5—figure supplement 1*). This indicates that like T$_H$17 differentiation, aerobic glycolysis promotes T cell growth by limiting supply of fructose-6-phosphate to the hexosamine pathway.

## Glutaminolysis drives T$_H$17 over iTreg differentiation by lowering N-glycan branching

Aerobic glycolysis is frequently accompanied by glutaminolysis, where glutamine is metabolized to glutamate by the enzyme glutaminase (*DeBerardinis et al., 2007*). GFPT1 absolutely requires glutamine as an amine donor to convert fructose-6-phosphate to glucosamine-6-phosphate, the first step of the de novo hexosamine pathway (*Figure 1A*). This suggests that glutaminolysis may reduce availability of glutamine to the hexosamine pathway, thereby lowering branching to drive T$_H$17 over iTreg differentiation. To examine this possibility, we investigated whether excess exogenous glutamine can reverse the reduction in branching induced by T$_H$17 cytokines and thereby drive iTreg over T$_H$17 differentiation. Indeed, under T$_H$17 conditions excess glutamine raised branching, markedly inhibited T$_H$17 differentiation and induced a cell fate switch to iTreg (*Figure 6A,B*, *Figure 6—figure supplement 1A*). These phenotypes were reversed by blocking branching with kifunensine, confirming that excess glutamine acted by raising branching. Similarly, increasing glutamine under iTreg-inducing conditions raised branching and increased iTreg generation in the absence but not in the presence of kifunensine (*Figure 6C,D*, *Figure 6—figure supplement 1B*). Blocking glutaminase activity with 6-diazo-5-oxo-L-norleucine (DON) or azaserine should increase glutamine supply to GFPT1. Indeed, both glutaminase inhibitors enhanced branching and induced a cell fate switch from T$_H$17 to iTreg (*Figure 6E–J*). Klysz et al. reported that glutamine deprivation drives iTreg over T$_H$1 differentiation by reducing glutamine catabolism to α-ketoglutarate, a conclusion suggested by the ability of the α-ketoglutarate analog dimethyl-2-oxoglutarate (DMK) to restore T$_H$1 over iTreg differentiation in the presence of glutamine deprivation (*Klysz et al., 2015*). In contrast, we observed that

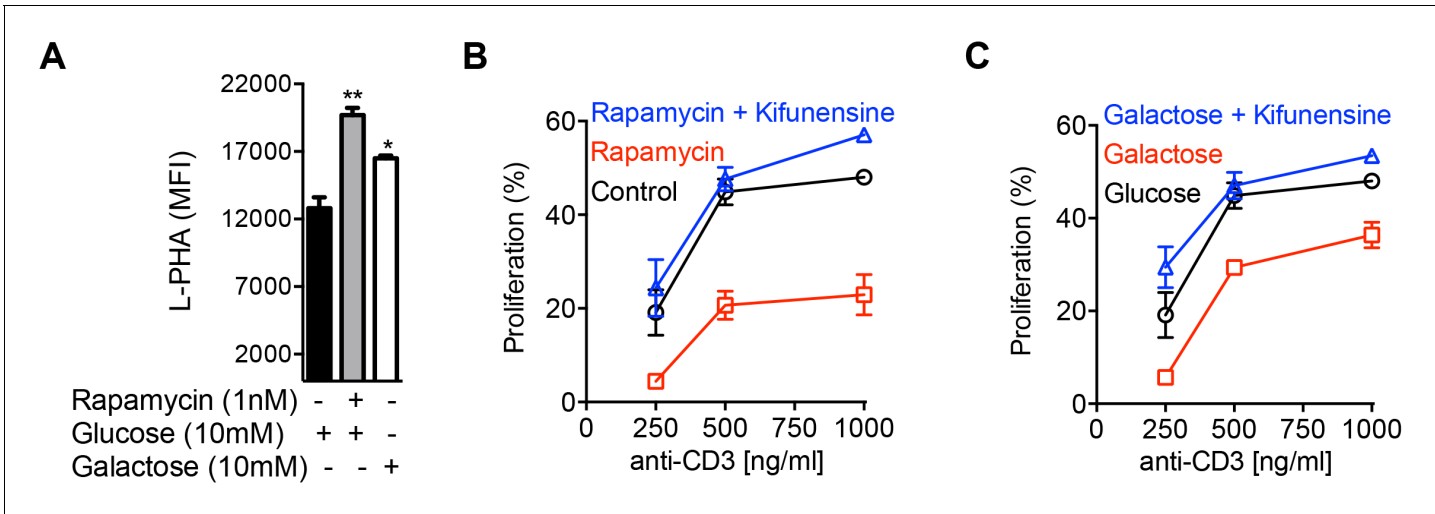

**Figure 5.** Glycolysis promotes T cell growth by inhibiting N-glycan branching. (**A–C**) Flow cytometry of purified splenic CD4$^+$ T cells activated with anti-CD3 for 3 days, gated on CD4$^+$. (**B,C**) Proliferation measured by CFSE dilution. *p<0.05; **p<0.01; ***p<0.001. (**A**) Unpaired two-tailed *t*-test with Bonferroni correction. Data are mean ± s.e.m of triplicate cultures and representative of n = 3 experiments. MFI, mean fluorescence intensity.

The following figure supplement is available for figure 5:

**Figure supplement 1.** Glycolysis promotes T cell growth by inhibiting N-glycan branching.

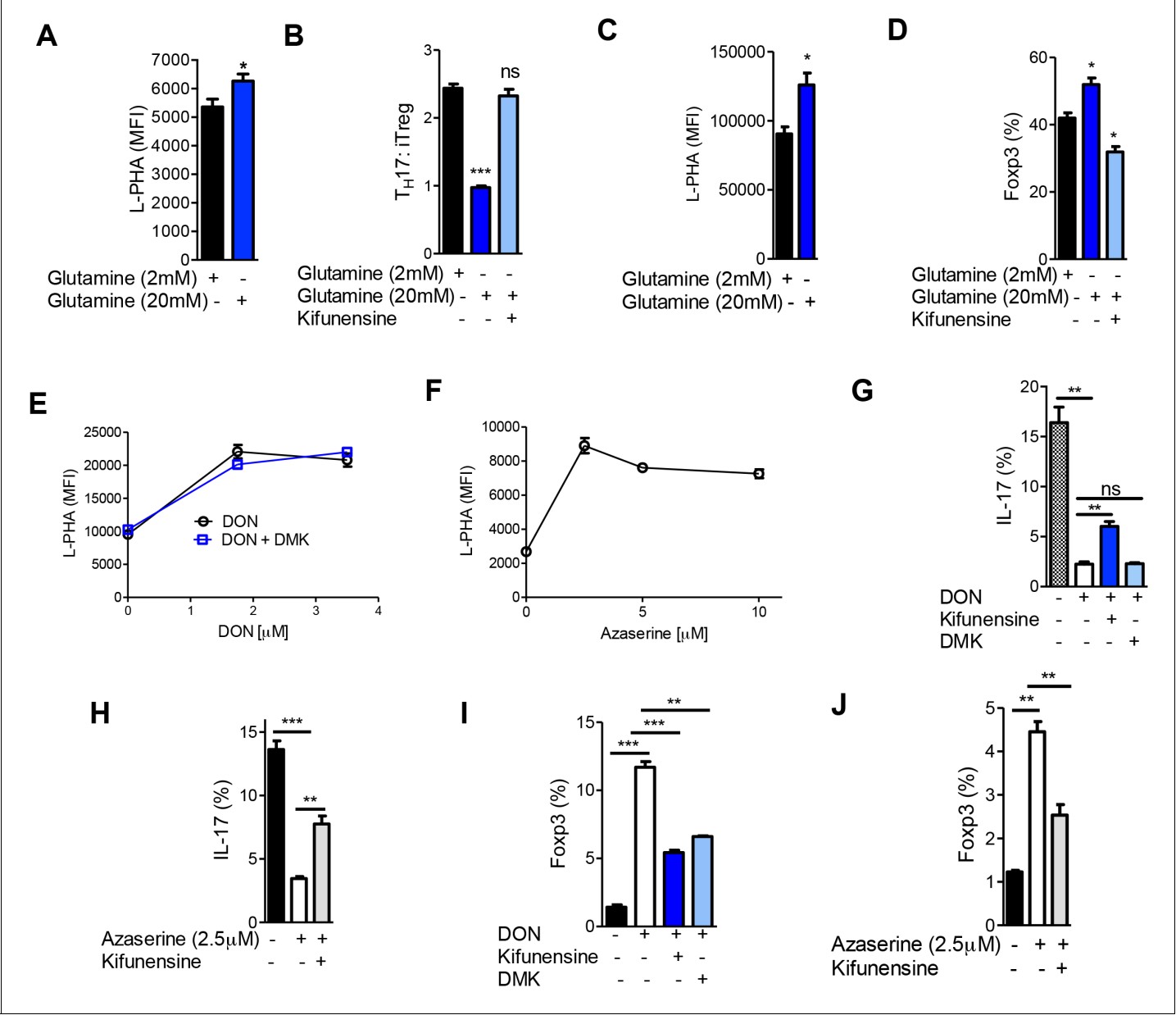

**Figure 6.** Glutaminolysis promotes T$_H$17 over iTreg cell fate by inhibiting N-glycan branching. (A–J) Flow cytometry of purified splenic CD4$^+$ T cells activated with anti-CD3+anti-CD28 under T$_H$17-inducing conditions (TGF$\beta$+IL-6+IL-23) (A,B,E–J) or in the presence of TGF$\beta$ (C,D). Gated on CD4$^+$. *p<0.05; **p<0.01; ***p<0.001. (A–D,G–J) Unpaired one-tailed *t*-test with Welch's (G,I,J) and Bonferroni (B,D,G–J) correction. Data are mean ± s.e.m and n = 3. MFI, mean fluorescence intensity.

The following figure supplement is available for figure 6:

**Figure supplement 1.** Glutaminolysis promotes T$_H$17 over iTreg cell fate by inhibiting N-glycan branching.

DMK did not reverse the effects of glutaminase inhibition by DON on branching or T$_H$17 induction (*Figure 6E,G*). However, DMK did induce a partial rescue of iTreg generation (*Figure 6I*). Moreover, blocking branching with kifunensine only partially reversed the effects of DON and Azaserine on T$_H$17 versus iTreg generation (*Figure 6G–J*). Together, these data indicate that glutamine availability regulates branching to affect T$_H$17 versus iTreg differentiation, but other factors such as catabolism of glutamine to $\alpha$-ketoglutarate likely also play a role.

## Discussion

It has been over 90 years since Otto Warburg's seminal discovery that cancer cells switch their metabolism from oxidative phosphorylation to aerobic glycolysis (*Warburg et al., 1927*; *Warburg et al., 1924*). Rapidly proliferating mammalian cells paradoxically stop the complete oxidation of glucose and instead ferment glucose to lactate while using glutamine rather than glucose to supply the TCA cycle. This extracts 18-fold less energy per glucose molecule while markedly increasing catabolism of glutamine, changes that should limit glucose and glutamine flux into other pathways. Indeed, our data support a model where a primary function of aerobic glycolysis and glutaminolysis is to co-operatively limit the flux of glucose and glutamine into the hexosamine pathway, thereby inhibiting N-glycan branching and associated down-stream phenotypes.

In T cells, reduced branching induced by aerobic glycolysis and glutaminolysis promotes proinflammatory $T_H17$ differentiation over anti-inflammatory iTreg differentiation. Restoring branching and UDP-GlcNAc levels via salvage of N-acetylglucosamine (GlcNAc) into the hexosamine pathway does not alter the glycolytic state of the cell, yet switches cell fate from $T_H17$ to iTreg cells by blocking endocytic loss of IL-2 receptor-$\alpha$ (CD25). Forcing oxidative phosphorylation raises branching to drive iTreg over $T_H17$ differentiation. Blocking the increase in branching reverses the latter phenotypes, demonstrating that aerobic glycolysis is not essential for T cell growth/differentiation control beyond regulation of branching. We conclude that metabolic switches between oxidative phosphorylation and aerobic glycolysis plus glutaminolysis primarily target hexosamine/N-glycosylation via altered glucose and glutamine flux and as a consequence, influence T cell differentiation and growth.

GlcNAc has recently been reported to promote glucose, glutamine, and fatty acid uptake in liver and/or adipose tissue via branching induced increases in surface retention of metabolite transporters (*Ryczko et al., 2016*). Similarly, we observed that modifying branching in T cells with GlcNAc or kifunensine altered glucose uptake. However, GlcNAc or kifunensine did not acutely alter the metabolic state of the T cell, as measured by the oxygen consumption rate and extracellular acidification rate. Consistent with this, we have previously shown that lowering branching does not alter UDP-GlcNAc levels in T cells (*Mkhikian et al., 2016*). GlcNAc does raise UDP-GlcNAc levels, which may impact other glycan pathways such as O-glycans, O-GlcNAc or CMP-sialic scid, in addition to N-glycan branching. However, the effects of GlcNAc on T cell differentiation were reversed by directly blocking Golgi branching activity, confirming that the effects of raised UDP-GlcNAc were predominantly through branching rather than other pathways. Moreover, blocking the effects of the metabolic modulators rapamycin, galactose or etoximer on branching reversed their effects on T cell differentiation despite not reversing effects on metabolism. In this manner, T cell differentiation can be decoupled from the metabolic state of the cell by manipulating branching. However, our data does not exclude metabolically driven changes in other pathways and indeed, the effects of glutaminolysis on branching appear to only partially explain the phenotype.

The competition between PFK1 and GFPT for fructose-6-phosphate is essential for determining the extent of glucose flux into the hexosamine/branching pathways. During $T_H17$ differentiation, GFPT1 is down-regulated and PFK1 dominates, restricting glucose flux into the hexosamine pathway and lowering branching relative to neutral T cell blasts. In contrast, GFPT1 is up-regulated in neutral T cell blasts, allowing increases in glucose uptake to raise de novo UDP-GlcNAc production and branching relative to resting cells (*Lau et al., 2007*); however, aerobic glycolysis still functions to limit the extent of glucose flux into the hexosamine pathway and thereby functions to promote growth.

The Warburg effect was first described in cancer cells, where branching and Mgat5 are pathologically over-expressed and serve as essential drivers of growth, motility and metastasis (*Granovsky et al., 2000*; *Lau et al., 2007*; *Partridge et al., 2004*). N-glycan branching and the hexosamine pathway promote epithelial to mesenchymal transition and tumor growth by regulating surface expression of receptor tyrosine kinases and TGF$\beta$ receptors (*Demetriou et al., 1995*; *Lau et al., 2007*; *Partridge et al., 2004*). While oncogenes such as Kras have been reported to turn on hexosamine and N-glycan pathway genes (*Ying et al., 2012*), the functional connection between aerobic glycolysis and hexosamine/branching in cancer has not been established. GFPT expression is elevated in a number of cancers (*Itkonen et al., 2013*; *Ying et al., 2012*) (R. Zhou and M. Demetriou, unpublished data), where it is expected to promote flux of glucose into the hexosamine pathway and branching. Our data in T cells suggest that although increased GFPT and MGAT5

expression in cancer drive UDP-GlcNAc and branching, aerobic glycolysis is expected to do the opposite and limit UDP-GlcNAc expression and branching. The latter would inhibit tumor growth and metastasis, but this phenotype may be masked by up-regulation of GFPT and/or MGAT5. This would parallel our results in neutral T cell blasts, where aerobic glycolysis limits branching yet elevated GFPT and MGAT5 act downstream to increase branching. The functional consequences of aerobic glycolysis and glutaminolysis on branching in cancer merit further investigation.

The ability of GlcNAc to promote iTreg while blocking $T_H17$ differentiation further validates its potential as a therapeutic for autoimmune disorders. We have previously shown that GlcNAc limits T cell activation/growth and when provided orally to mice, inhibits experimental autoimmune encephalomyelitis, a mouse model of Multiple Sclerosis (MS), as well as autoimmune diabetes in the Non Obese Diabetic mouse model (*Grigorian et al., 2011*, *2007*). GlcNAc has also been given orally (3–6 g/day) to children with refractory inflammatory bowel disease for ~2 years, with 8 of 12 showing clinical improvement without reported toxicities and/or side effects (*Salvatore et al., 2000*). We have recently observed that serum levels of endogenous GlcNAc are markedly reduced in patients with the progressive form of MS and correlate with clinical disability and imaging measures of neurodegeneration (Alexander Brandt and Michael Demetriou, unpublished data). A pilot study of low-dose oral GlcNAc in MS (3 g/day) increased serum GlcNAc levels and branching in T cells (Barbara Newton and Michael Demetriou, unpublished data). As GlcNAc is a dietary supplement that is for sale 'over the counter' in the US, these data suggest that GlcNAc may serve as a safe and inexpensive therapeutic for MS patients and potentially other autoimmune diseases.

In summary, our data brings new clarity to the function of aerobic glycolysis and glutaminolysis and has widespread implications for the role of metabolism in disease states as diverse as autoimmunity and cancer.

## Materials and methods

### Mice

*Mgat5*-deficient and doxycycline-inducible *Mgat1*-deficient mice (*Mgat1*$^{f/f}$tetO-Cre$^+$ROSA$^{rtTA}$) were previously described (*Demetriou et al., 2001*; *Zhou et al., 2014*). To delete *Mgat1* in peripheral T cells, 2 mg/ml doxycycline plus 1% sucrose was provided in the drinking water to *Mgat1*$^{f/f}$tetO-Cre$^+$-ROSA$^{rtTA}$ mice for $\geq 3$ weeks. To generate doxycycline-inducible *Mgat5* over-expression mice (*Mgat5*$^{tetO}$ROSA$^{rtTA}$), the cDNA encoding the mouse *Mgat5* (a gift from James Dennis) was subcloned into the transgene expression vector pTRE-Tight (Clontech, Mountain View, CA), containing a tet-regulated promoter controlling the expression of the subcloned *Mgat5* cDNA (*Mgat5*$^{tetO}$). The transgene was microinjected into fertilized C57BL/6 mouse eggs and placed into pseudo-pregnant females. Founder mice were bred with C57BL/6 mice and transgenic offspring were identified by PCR using a PCR primer pair targeting the Tet promoter and *Mgat5* transgene (left – CGAGG TAGGCGTGTACGG, right – AGACCGGTTTCCAACAACCT). Transgenic-positive mice were then crossed with ROSA$^{rtTA}$ to generate *Mgat5*$^{tetO}$ROSA$^{rtTA}$ mice and detected using primers as per Jackson Laboratory (stock 006965). To overexpress Mgat5 in mouse splenic CD4$^+$ T cells, doxycycline was added to cell culture at 4 µg/ml. All the mice were on the C57BL/6 background.

### Mouse T-cell cultures

For $T_H17$ cell induction, purified splenic CD4$^+$ T cells (EasySep Mouse CD4$^+$ T cell isolation kit, StemCell Technologies) were cultured for 4 days with plate-bound anti-CD3ε (0.25–5 µg/ml, clone 145–2 C11, eBioscience, San Diego, CA), anti-CD28 (2 µg/ml, clone 37.51, eBioscience), recombinant mouse IL-6 (20 ng/ml, eBioscience), recombinant human TGF-β1 (5 ng/ml, eBioscience), recombinant mouse IL-23 (20 ng/ml, eBioscience), anti-IFNγ (10 µg/ml, clone XMG1.2, eBioscience) and anti IL-4 (10 µg/ml, clone 11B11, eBioscience). For iTreg cell induction, purified splenic CD4$^+$ T cells were cultured for 4 days in the presence of plate-bound anti-CD3ε (1 µg/ml), anti-CD28 (2 µg/ml) and rhTGF-β1 (0.625–10 ng/ml). For pharmacological treatments, cells were incubated with vehicle, GlcNAc (40 mM added daily, Wellesley Therapeutics, Toronto, Canada), kifunensine (10 uM, Glycosyn, Wellington, New Zealand), rapamycin (1 nM, EMD Milipore, Temecula, CA), 2-Deoxy-D-glucose (1 mM, Sigma-Aldrich, St Louis, MO), Etomoxir ((Etx)200 µM, Sigma-Aldrich), 2-NBDG (20 µM, ThermoFisher Scientific, Waltham, MA), azaserine (2.5–10 µM, Sigma), 6-diazo-5-oxo-L-norleucine ((DON)

1.75–3.5 µM, Sigma), dimethyl-2-oxoglutarate ((DMK 3.5 mM, Sigma), anti-IL-2 (20 µg/ml eBioscience), Etomoxir and IL-2 (eBioscience). Cytokine production in cell culture supernatants was analyzed by enzyme-linked immunosorbent assay (ELISA) with mouse Ready Set Go! IL-2 assay kits (eBioscience) according to the manufacturer's instructions.

## Western blot

Western blotting was performed as previously described (*Demetriou et al., 2001*; *Grigorian et al., 2007*; *Zhou et al., 2014*). Cells were stimulated by plate-bound anti-CD3ε plus anti-CD28 for 72 hr before lysis. Un-stimulated cells were rested in culture for 12 hr prior to cell lysis. Whole-cell lysates were prepared as previously described (*Demetriou et al., 2001*; *Grigorian et al., 2007*; *Zhou et al., 2014*) and supplemented with Halt Protease Phosphotase Inhibitor (Fisher). Cell lysates were separated by SDS-gel electrophoresis and transferred to nitrocellulose membranes (ThermoFisher Scientific). Antibodies to Hexokinase I (Clone C3C54), phospho-Stat5 (Clone D47E7), PFK-L (Catalog #8175), actin (Clone 13E5), Hsp90 (Clone C45G5) and HRP-conjugated rabbit IgG (Catalog #7074) were from Cell Signaling Technology (Danvers, MA). Antibodies to PFK-P (Catalog #PA5-28673) and GPI (Catalog #PA5-26787) were from ThermoFisher Scientific. PFK-M (Clone 842735) was from R&D Systems (Minneapolos, MN). GFPT1 (Clone EPR4854) was from LifeSpan Biosciences (Seattle, WA). Hexokinase I, PFKP, PFKL, PFKM, anti-GFPT1, anti-GPI abundance were calculated by normalization to actin or Hsp90 and relative to control using Image J software as previously described (*Zhou et al., 2014*).

## Analysis of cell metabolism

Cells were harvested after 48 hr of culture and plated on 24-well XF cell culture micro-plates coated with Cell-Tak (Corning, Corning, NY) at a concentration of 22.4 µg/ml in non-buffered RPMI 1640 (containing either 10 mM glucose or 10 mM galactose, 2 mM L-glutamine and 1 mM sodium pyruvate) (Sigma) at a density of $7.5 \times 10^5$ cells per well. Using two-step plating, microplates were incubated for 10 min with 100 µl of cells in non-buffered media at 37°C in a non-CO$_2$ incubator. Then, 400 µl of non-buffered media was added to each well and incubated for an additional 20 min. The Extracellular Acidification Rate (ECAR) and Oxygen Consumption Rate (OCR) were measured using an XF24 Extracellular Flux Analyzer (Seahorse Biosciences/Agilent Technologies, Santa Clara, CA) under basal conditions.

## Flow cytometry and intracellular staining

Flow cytometric analysis was performed as previously described (*Demetriou et al., 2001*; *Grigorian et al., 2007*; *Zhou et al., 2014*). Monoclonal antibodies specific to the following mouse antigens (and labeled with the indicated fluorescent markers) were purchased from eBioscience: IL-17 PE (eBio17B7), Foxp3 APC (Fjk-16s), CD4 PrpCy5.5 (RM4-5), CD25 PE (PC61.5). In addition, FITC L-PHA, FITC E-PHA and FITC ConA (Vector Labs, Burlingame, CA) was used to stain cells. For galectin-3 binding, recombinant mouse galectin-3 (R&D systems) was labeled using an Alexa Fluor 488 protein labeling kit (ThermoFisher Scientific). Cells were stained for flow cytometry using 3 µg labeled galectin-3 per test. Staining was carried out for 30 min at room temperature followed by one wash and fixation with paraformaldehyde. For the lactose control samples, 50 mM lactose was included in all steps. For intracellular staining, cells were stimulated at 37°C for 5 hr with phorbol 12-myristate 13-acetate (50 ng/ml, Sigma-Aldrich) and ionomycin (750 ng/ml, Sigma-Aldrich) in the presence of GolgiPlug (1 µl per 1 ml cell culture) (BD Biosciences) and stained using the Foxp3 Transcription Factor Fixation/Permeabilization Kit (eBioscience) according to the manufacturer's instruction. Flow cytometry experiments were performed with BD FACS Calibur, LSR II and Attune Acoustic Focusing Cytometer.

## Endocytosis assay

Endocytosis assay was assessed as previously described (*Mkhikian et al., 2011*; *Zhou et al., 2014*). Briefly, splenocytes were stained with anti-CD25, resuspended in complete RPMI 1640 medium, incubated in a 37°C incubator for 2.5 hr, washed in FACS buffer or acid-wash buffer (150 mM NaCl and 20 mM HCl, pH 1.7) for 3 min at 25°C and then fixed before analyzing by flow cytometry. The

acid wash removes surface-bound antibody and the MFI of acid-washed cells is divided by the MFI of FACS buffer washed cells to determine internalized antibody.

## UDP-GlcNAc measurement by LC-MS/MS

Purified splenic CD4$^+$ T cells activated for 72 hr in the presence or absence of T$_H$17-inducing conditions were washed twice with ice-cold 1x PBS and counted. $15 \times 10^6$ cells were pelleted in 1.5 ml Eppendorf tubes by centrifuging at 350 g for 7 min at 4°C. Metabolites were extracted from the pellets by the addition of 1 ml of ice-cold solution of 40% acetonitrile, 40% methanol and 20% water. The tubes were vortexed for 1 hr at 4°C and spun down at 14,000 rpm for 10 min at 4°C (Eppendorf, Germany). The supernatant was transferred to fresh tubes and evaporated to dryness in an Eppendorf Vacufuge at 30°C (Eppendorf, Germany). The dry samples were stored at −80°C until analyzed, at which point they were reconstituted in 100 µl of LC-MS grade water. The samples were run on a Waters (Micromass) Quattro Premier XE LC-MS/MS machine in negative mode with Water:Acetonitrile (+0.2% Acetic Acid and 5 mM Ammonium Acetate) solvent. UPLC with C18 reverse phase column was employed. Area was converted to UDP-GlcNAc concentration using a standard curve simultaneously generated with purified UDP-GlcNAc (Sigma).

## Acknowledgements

Research was supported by R01AI053331 and R01AI108917 from the National Institute of Allergy and Infectious Disease and R01 AT007452 from the National Center for Complementary and Integrative Health to MD. We thank Yumay Chen of the Ping Wang lab for helping with experiments and Barbara Newton for proofreading the manuscript.

## Additional information

### Funding

| Funder | Grant reference number | Author |
| --- | --- | --- |
| National Institute of Allergy and Infectious Diseases | R01 AI053331 | Michael Demetriou |
| National Center for Complementary and Integrative Health | R01 AT007452 | Michael Demetriou |
| National Institute of Allergy and Infectious Diseases | R01 AI108917 | Michael Demetriou |

The funders had no role in study design, data collection and interpretation, or the decision to submit the work for publication.

### Author contributions

LA, Conceptualization, Formal analysis, Investigation, Methodology, Writing—original draft; PK, C-LM, Data curation, Investigation; HM, Data curation, Formal analysis, Investigation, Writing—review and editing; MD, Conceptualization, Resources, Formal analysis, Supervision, Funding acquisition, Investigation, Methodology, Writing—original draft, Project administration, Writing—review and editing

### Author ORCIDs

Michael Demetriou, http://orcid.org/0000-0001-8547-5774

### Ethics

Animal experimentation: This study was performed in strict accordance with the recommendations in the Guide for the Care and Use of Laboratory Animals of the National Institutes of Health. All of the animals were handled according to approved institutional animal care and use committee (IACUC) protocols (#2001-2305) of the University of California, Irvine.

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
