## [Decision Letter]

Thank you for submitting your article "Glycolysis and glutaminolysis cooperatively control T cell function by limiting metabolite supply to N-glycosylation" for consideration by *eLife*. Your article has been favorably evaluated by Michel Nussenzweig (Senior Editor) and three reviewers, one of whom served as Guest Reviewing Editor. The reviewers have opted to remain anonymous.

The reviewers have discussed the reviews with one another and the Reviewing Editor has drafted this decision to help you prepare a revised submission.

General assessment:

This interesting report shows that pro-inflammatory cytokines induce a metabolic switch in T cells to aerobic glycolysis and glutaminolysis that starves the hexosamine pathway, reduces UDP-GlcNAc levels, and as a consequence reduces branching of N-linked glycans that coincides with the differentiation to pro-inflammatory T_H_17 cells and suppression of iTregs. While the data provide significant insights into the relationship between induction of aerobic catalysis and changes in glycosylation, the reviewers felt that the report implies a causal relationship between branching of N-linked glycans and differentiation of T cells to iTregs that is not fully supported by the data.

Major conclusions:

– T_H_17 cytokines induces a metabolic switch to aerobic glycolysis, differentiation of pro-inflammatory T_H_17 cells, and suppression of differentiation to iTreg cells.

– Aerobic glycolysis and reduced glutaminolysis consumes precursors of GlcNAc and UDP-GlcNAc, resulting in reduced branching of N-linked cytokines.

– Bypassing the suppression of N-linked glycan branching in the presence of T_H_17 cytokines by administering GlcNAc or, to a lesser extent, by increasing the expression of the branching enzyme, Mgat5, restores branching of N-linked glycans and reverses suppression of differentiation to iTregs.

– Causal relationships between N-linked glycan branching and function of surface receptors (e.g. IL2 receptor, CD25) suggest that N-linked glycan branching may play a direct role in driving T_H_17/ iTreg differentiation.

Required revisions:

The reviewers expressed major concerns about narrowly focusing interpretation of the results, with a bias that N-linked glycan branching controlled T cell differentiation. This was to the exclusion of the potential for branching to be coincidental to factors that induced differentiation or even other aspects of glycosylation that could be playing a principle role (e.g. O-GlcNAc glycosylation). A related concern was the lack of data confirming that changes in N-linked branching was taking place. Changes were only measured with the lectin PHA-L. There was no effort to assess how quantitative the changes in branching were. Addressing these concerns should be a priority in a revised manuscript.

There are several places in the manuscript that suggest the metabolic switch to aerobic glycolysis targets the hexosamine/N-glycosylation pathway to control T cell differentiation. Rather what has been shown is that the downstream effects are a consequence of the switch to aerobic glycolysis. The implication that this metabolic switch occurs to control these events goes beyond what is shown. It is a subtle difference but an important one.

Abstract, bottom line: “and N-glycan branching to control cell function” should be “N-glycan branching that controls cell function”.

Discussion, first paragraph: “aerobic glycolysis and glutaminolysis act in concert to control cell growth and differentiation”.

Discussion, second paragraph, last sentence: “We conclude that metabolic switches… target hexosamine/N-glycosylation… to control T cell differentiation and growth”.

The authors ignore a recent article (PMID 26972830) reporting that providing GlcNAc to mice "increased N-glycan branching" and showed altered glucose homeostasis (increased uptake), hepatic glycogen, lipid metabolism, and response to fasting. In cultured cells they reported "enhanced uptake of glucose, glutamine, and fatty acids" and "enhanced lipid synthesis". These results show that GlcNAc supplementation does not only impact N-glycan branching. It would change N- and O-linked glycan structures (not measured here). GlcNAc can be also epimerized to ManNAc and fuel the sialic acid biosynthetic pathway (not measured here). GlcNAc supplementation and the concomitant increase in glucose uptake would significantly impact O-GlcNAc levels, which alters many cell biological processes, including those of T cell differentiation (not explored here). In other parts of the article they supplement with galactose, but this is known to increase sialylation (PMID 25931375), and supplementation with glucose (also performed in this study) drives O-GlcNAcylation.

The authors have chosen to look at only one type of structure (N-glycan branching, as measured by PHA-L,) and never performed complete glycan composition from the cell surface, an overall measure of O-GlcNAc levels in cytoplasmic and nuclear proteins, or any detailed glycomic analysis.

Mgat5 is claimed to be responsible for the synthesis of glycan branch recognized by PHA-L by the authors. The deletion of Mgat5 in Figure 4 has only a modest change in the induction of Tregs, yet these cells should be completely PHA-L negative. Compare this to Figure 6. The small change in PHA-L (Figure 6) is not convincingly associated with the large reported change in T cell phenotype in Figure 6, especially given the magnitudes shown in Figure 4 where PHA-L staining should be gone. This suggests that something not measurable with PHA-L is at play and that PHA-L is a poor measure of outcome.

mTOR is O-GlcNAcylated (PMID 20410138). Altering the balance of UDP-GlcNAc through various means in a cell, including the use of kifunensine, could easily alter the function of mTOR. The authors do not explore this possibility.

Several comments about changes in receptor function resulting from N-linked glycan branching should be addressed:

IL-2 is required for Treg survival, thus the conclusion reached by the authors that IL-2 "is required for branching to promote iTreg over T_H_17 differentiation (Figure 2)" is not directly supported by the data. Without IL-2, Tregs die, with or without highly branched glycans.

The authors conclude that branching drives CD25 surface retention, but they only measure CD25 surface concentration. There is no direct data showing anything about surface retention. It is possible that the expression is merely lower. It is possible that the Mgat1 mutation creates problems with protein folding (a common effect). It is possible that it never even reaches the cell surface.

For readers not familiar with the Warburg effect and aerobic glycolysis, it would be helpful to mention that glycolysis is the conditional norm of anaerobic metabolism, and that the discovery by Warburg was that cancer cells constitutively use glycolysis even in aerobic conditions (e.g. aerobic glycolysis).

---

## [Author Response]

*General assessment:*

*This interesting report shows that pro-inflammatory cytokines induce a metabolic switch in T cells to aerobic glycolysis and glutaminolysis that starves the hexosamine pathway, reduces UDP-GlcNAc levels, and as a consequence reduces branching of N-linked glycans that coincides with the differentiation to pro-inflammatory T_H_17 cells and suppression of iTregs. While the data provide significant insights into the relationship between induction of aerobic catalysis and changes in glycosylation, the reviewers felt that the report implies a causal relationship between branching of N-linked glycans and differentiation of T cells to iTregs that is not fully supported by the data.*

*Major conclusions:*

*– T_H_17 cytokines induces a metabolic switch to aerobic glycolysis, differentiation of pro-inflammatory T_H_17 cells, and suppression of differentiation to iTreg cells.*

*– Aerobic glycolysis and reduced glutaminolysis consumes precursors of GlcNAc and UDP-GlcNAc, resulting in reduced branching of N-linked cytokines.*

*– Bypassing the suppression of N-linked glycan branching in the presence of T_H_17 cytokines by administering GlcNAc or, to a lesser extent, by increasing the expression of the branching enzyme, Mgat5, restores branching of N-linked glycans and reverses suppression of differentiation to iTregs.*

*– Causal relationships between N-linked glycan branching and function of surface receptors (e.g. IL2 receptor, CD25) suggest that N-linked glycan branching may play a direct role in driving T_H_17/ iTreg differentiation.*

*Required revisions:*

*The reviewers expressed major concerns about narrowly focusing interpretation of the results, with a bias that N-linked glycan branching controlled T cell differentiation. This was to the exclusion of the potential for branching to be coincidental to factors that induced differentiation or even other aspects of glycosylation that could be playing a principle role (e.g. O-GlcNAc glycosylation).*

The reviewers raise the major issue that our data only supports the conclusion that alterations in branching induced by metabolic changes are coincidental rather than causal for T cell differentiation. We fully agree that by itself, our data demonstrating that GlcNAc raised UDP-GlcNAc/branching and affected T cell differentiation may be coincidental to other factors altered by metabolism, including other glycans (e.g. O-GlcNAc) that we did not assess. However, we feel our data goes well beyond this correlation to provide strong evidence of a causal relationship:

First, we show that directly altering branching enzyme activity in the Golgi with either a small molecule (kifunensine) or three different genetic approaches (inducible knockout of Mgat1, knockout of Mgat5 and inducible transgenic over-expression of Mgat5) all directly alter T_H_17 and iTreg differentiation (see Figure 1; Figure 1—figure supplement 1; Figure 1—figure supplement 2; Figure 4).

Second, we have previously published in *eLife* that inhibition of branching with small molecules (kifunensine, swainsonine) or genetically (Mgat2 deficiency) does not alter UDP-GlcNAc levels in T cells (Mkhikian et al. (2016) *eLife* 5 e14814). Therefore, altering branching should have no secondary effects on other UDP-GlcNAc dependent pathways, such as O-GlcNAc.

Third, we show that the effects of GlcNAc on T cell differentiation are reversed by directly blocking Golgi branching activity with kifunensine (Figure 1—figure supplement 1, Figure 4). If other pathways were essential (e.g. O-GlcNAc), then blocking branching would not have reversed the effects of GlcNAc.

Fourth, we show that GlcNAc and kifunensine do not significantly alter the metabolic state of the T cell (as measured by the Oxygen Consumption Rate versus the Extracellular Acidification Rate – see Figure 3), yet result in profound changes in T cell differentiation. Thus, manipulating branching does not acutely induce significant metabolic changes in the T cell, consistent with our published data that reducing branching does not alter UDP-GlcNAc levels in T cells (Mkhikian et al. (2016) *eLife* 5 e14814). Note that we do show that altering branching with GlcNAc and kifunensine induces a mild change in glucose uptake (Figure 3—figure supplement 1), but this is insufficient to significantly alter aerobic glycolysis/oxidative phosphorylation or UDP-GlcNAc levels in T cells.

Fifth, we demonstrate that T_H_17 versus iTreg differentiation can be de-coupled from the metabolic state of the cell by manipulating branching. Specifically, we show that inhibiting aerobic glycolysis with rapamycin or galactose as well as oxidative phosphorylation with Etoxomir not only alters branching as predicted (Figure 3, Figure 4, Figure 5), but that reversing the changes in branching (using GlcNAc, kifunensine and/or inducible Mgat1 KO) blocks the effects of the metabolic inhibitors on T cell differentiation/proliferation (Figure 3; Figure 3—figure supplement 1; Figure 4, Figure 5). Most importantly, blocking branching with kifunensine did not reverse the inhibition of aerobic glycolysis induced by rapamycin or galactose (Figure 3), yet reversed their effects on T_H_17 versus iTreg differentiation (Figure 3; Figure 3—figure supplement 1). This demonstrates that the change in branching induced by altering metabolism is not simply co-incidental but required for altering T_H_17 versus iTreg differentiation.

Thus, while we fully agree with the reviewers that metabolic changes in the cell and GlcNAc likely alter other pathways, our data provide strong evidence that metabolically driven changes in branching are *causal* in controlling T_H_17 versus iTreg differentiation. However, we agree that this does not exclude a role for other pathways. Indeed, we observed that the effects of glutaminolysis on branching only partially explain the effects on T cell differentiation. This was based on our observation that 1) while the α-ketoglutarate analog DMK did not reverse the effects of glutaminase inhibition by DON on branching or T_H_17 differentiation (Figure 6), it did partially rescue effects on iTreg differentiation (Figure 6) and 2) blocking branching with kifunensine only partially reversed the effects of glutaminase inhibition by DON and Azaserine on T_H_17 versus iTreg generation (Figure 6). These data are discussed in the last few sentences of the Results section, where we state “Together, these data indicate that glutamine availability regulates branching to affect T_H_17 versus iTreg differentiation, but other factors such as catabolism of glutamine to α-ketoglutarate likely also play a role.” We have also added a paragraph to the Discussion to further emphasize these points.

*A related concern was the lack of data confirming that changes in N-linked branching was taking place. Changes were only measured with the lectin PHA-L. There was no effort to assess how quantitative the changes in branching were. Addressing these concerns should be a priority in a revised manuscript.*

Thank you for this suggestion. To confirm reduction in N-glycan branching, we have added new flow cytometry data with additional lectins demonstrating that T_H_17 cytokines 1) reduced LacNAc content (i.e. reduced Galectin-3 binding), 2) reduced bisecting branched N-glycans (i.e. reduced E-PHA binding) and 3) increased high-mannose N-glycans (i.e. increased ConA binding) (Figure 1—figure supplement 1). All of these changes will lead to reduced galectin binding to N-glycans, thereby weakening the galectin-glycoprotein lattice and drive downstream phenotypes.

In terms of quantitation, measuring ‘mean fluorescence intensity’ by flow cytometry is the primary method to *quantitatively* assess levels of molecules at the cell surface. Others and we have previously validated flow cytometry with L-PHA as a highly sensitive, specific and quantitative measure of branching at the cell surface (via comparison to MALDI-TOF mass spectroscopy and HPEAC – see Lau et al. (2007) Cell, Grigorian et al. (2007, 2011) JBC and Mkhikian et al. (2016) *eLife*). As the enzymes in the N-glycan branching pathway act linearly and with declining efficiency, production of β1,6 GlcNAc branched tri- and tetra-antennary N-glycans (the ligand for L-PHA) not only depends on Mgat5 activity but also on the activity of upstream enzymes (e.g. Mgat1, Mgat2, mannosidase I, mannosidase II). For example, titrations of inhibitors for mannosidase I (kifunensine) or mannosidase II (swainsonine) show a direct relationship to changes in L-PHA binding as measured by flow cytometry. Thus, L-PHA flow cytometry is a quantitative measure of the overall output of the branching pathway, rather than simply a measure of Mgat5 activity.

Point by point responses to other issues are presented below.

There are several places in the manuscript that suggest the metabolic switch to aerobic glycolysis targets the hexosamine/N-glycosylation pathway to control T cell differentiation. Rather what has been shown is that the downstream effects are a consequence of the switch to aerobic glycolysis. The implication that this metabolic switch occurs to control these events goes beyond what is shown. It is a subtle difference but an important one.

Abstract, bottom line: “and N-glycan branching to control cell function” should be “N-glycan branching that controls cell function”.

The suggested change has been made (last sentence of Introduction).

*Discussion, first paragraph: “aerobic glycolysis and glutaminolysis act in concert to control cell growth and differentiation”.*

The sentence has been deleted.

*Discussion, second paragraph, last sentence: “We conclude that metabolic switches… target hexosamine/N-glycosylation… to control T cell differentiation and growth”.*

This has been changed to “We conclude that metabolic switches between oxidative phosphorylation and aerobic glycolysis plus glutaminolysis primarily target hexosamine/N-glycosylation via altered glucose and glutamine flux, and as a consequence influence T cell differentiation and growth.”

*The authors ignore a recent article (PMID 26972830) reporting that providing GlcNAc to mice "increased N-glycan branching" and showed altered glucose homeostasis (increased uptake), hepatic glycogen, lipid metabolism, and response to fasting. In cultured cells they reported "enhanced uptake of glucose, glutamine, and fatty acids" and "enhanced lipid synthesis". These results show that GlcNAc supplementation does not only impact N-glycan branching. It would change N- and O-linked glycan structures (not measured here). GlcNAc can be also epimerized to ManNAc and fuel the sialic acid biosynthetic pathway (not measured here). GlcNAc supplementation and the concomitant increase in glucose uptake would significantly impact O-GlcNAc levels, which alters many cell biological processes, including those of T cell differentiation (not explored here). In other parts of the article they supplement with galactose, but this is known to increase sialylation (PMID 25931375), and supplementation with glucose (also performed in this study) drives O-GlcNAcylation.*

PMID 26972830 and our work differ in that PMID 26972830 is assessing the downstream effects of GlcNAc/branching on metabolism while we do the opposite, examining the downstream effects of metabolism on branching. Specifically, PMID 26972830 argues that GlcNAc is promoting branching, which in turn affects metabolite uptake in the liver and adipose tissue via branching enhancing surface retention of metabolite transporters. In contrast, we are assessing the effects of aerobic glycolysis/oxidative phosphorylation on branching and associated downstream phenotypes in T cells. We use GlcNAc salvage into the hexosamine pathway as a mechanism to by-pass metabolic effects of aerobic glycolysis on the de novo hexosamine pathway. Importantly, and in contrast to PMID 26972830, we show that neither GlcNAc or kifunensine acutely altered the metabolic state of the T cell (as measured by the oxygen consumption rate and extracellular acidification rate – Figure 3). This was despite altering glucose uptake (Figure 3—figure supplement 1), as observed in PMID 26972830. Moreover, we have previously shown that lowering branching does not alter UDP-GlcNAc levels in T cells (Mkhikian et al. (2016) *eLife* 5 e14814) and therefore our experiments inhibiting Golgi branching activity should have no effect on O-GlcNAc or CMP-Sialic Acid production. Thus, our observation that T cell differentiation is changed by directly altering branching activity in the Golgi (e.g. kifunensine, Mgat1 KO, Mgat5 KO and inducible Mgat5 over-expression) demonstrates causality of branching on T_H_17 versus iTreg differentiation.

GlcNAc does raise UDP-GlcNAc levels, which in theory may impact O-GlcNAc or CMP-Sialic Acid in addition to branching. However, we show that the effects of GlcNAc on T cell differentiation are reversed by directly blocking Golgi branching activity (Figure 1—figure supplement 1, Figure 4), confirming that the effects of raised UDP-GlcNAc were predominantly through branching rather than other pathways. Most importantly, we show that blocking the effects of the metabolic modulators rapamycin, galactose or etoximer on branching reversed their effects on T cell differentiation (Figure 3; Figure 3—figure supplement 1; Figure 4, Figure 5) despite not reversing effects on metabolism (Figure 3). In this manner, we fully decoupled T cell differentiation from the metabolic state of the cell by manipulating branching. Combined, this data demonstrates that metabolism induced changes in N-glycan branching are causal in regulating T cell differentiation. However, we acknowledge that this does not exclude metabolically driven changes in other pathways and indeed we state that the effects of glutaminolysis on branching only partially explain the phenotype (see last few sentences of Results section). A summary of the above arguments has been added to the Discussion.

*The authors have chosen to look at only one type of structure (N-glycan branching, as measured by PHA-L,) and never performed complete glycan composition from the cell surface, an overall measure of O-GlcNAc levels in cytoplasmic and nuclear proteins, or any detailed glycomic analysis.*

As noted above, we confirmed T_H_17 cytokine induced changes in N-glycan branching using flow cytometry with 3 additional lectins. Also, note that we have previously utilized mass spec and HPEAC to validate changes in N-glycans observed with lectins.

Our first fundamental observation was that T_H_17 cytokines markedly reduced branching in T cells. We then demonstrated that directly altering Golgi branching enzyme activity markedly affected T_H_17 versus iTeg differentiation. Thus, the next logical step was to evaluate what the mechanism for altered branching during T cell differentiation. As T_H_17 cells are highly glycolytic while iTreg cells utilize lipid oxidation, we pursued the hypothesis that the changes in branching were metabolically driven. Our goal was not to broadly assess downstream effects of metabolism. As we go on to demonstrate a causal link between aerobic glycolysis/oxidative phosphorylation, branching and T cell differentiation (see discussion above), assessing other pathways is beyond the scope of this investigation.

*Mgat5 is claimed to be responsible for the synthesis of glycan branch recognized by PHA-L by the authors. The deletion of Mgat5 in Figure 4 has only a modest change in the induction of Tregs, yet these cells should be completely PHA-L negative. Compare this to Figure 6. The small change in PHA-L (Figure 6) is not convincingly associated with the large reported change in T cell phenotype in Figure 6, especially given the magnitudes shown in Figure 4 where PHA-L staining should be gone. This suggests that something not measurable with PHA-L is at play and that PHA-L is a poor measure of outcome.*

As the enzymes in the N-glycan branching pathway act linearly and with declining efficiency, production of β1,6 GlcNAc branched tri- and tetra-antennary N-glycans that L-PHA binds not only depends on Mgat5 activity but also on the activity of upstream enzymes (e.g. Mgat1, Mgat2, mannosidase I, mannosidase II). In this manner, L-PHA flow cytometry of wildtype cells is a quantitative measure of the overall output of the branching pathway, rather than simply a measure of Mgat5 activity. Thus, the small increase in L-PHA binding in 6A reflects not only increased β1,6 GlcNAc branching by Mgat5, but also increased branching by Mgat1 and Mgat2. Importantly, the relationship between L-PHA binding and overall branching is not true when examining genetically targeted cells deleted for branching enzymes. For example, L-PHA does not bind Mgat5^-/-^ or Mgat1^-/-^ T cells, yet the former only reduces galectin-3 binding ~50% while the latter eliminates all LacNAc ligands for galectins (Mkhikian et al. (2016) *eLife* 5 e14814 and Zhou et al. (2014) Nature Immunology). Thus, Mgat1 deletion is a much more severe disruption of the galectin – glycoprotein lattice than Mgat5 deletion even though both lack L-PHA binding. This explains why phenotypic changes are more severe in the Mgat1 versus Mgat5 KO cells (e.g. compare Figure 4 to 4B).

Also, note that we state that the effects of glutaminolysis on branching only partially explain the phenotypic effects of glutaminolysis (see last few sentences of Results section).

*mTOR is O-GlcNAcylated (PMID 20410138). Altering the balance of UDP-GlcNAc through various means in a cell, including the use of kifunensine, could easily alter the function of mTOR. The authors do not explore this possibility.*

As noted above, total cellular UDP-GlcNAc levels are not altered by kifunensine or deletion of branching enzymes in T cells (Mkhikian et al. (2016) *eLife* 5 e14814). Moreover, mTOR drives glycolysis (via HIF1) yet neither kifunensine or GlcNAc significantly altered the glycolytic state of the T cell (Figure 3). Thus, there is little reason to expect that the manipulations of branching will affect O-GlcNAcylation of mTOR. In contrast, we do show that mTOR has profound effects on branching, as rapamycin markedly enhanced L-PHA binding in T cells (Figure 3).

*Several comments about changes in receptor function resulting from N-linked glycan branching should be addressed:*

*IL-2 is required for Treg survival, thus the conclusion reached by the authors that IL-2 "is required for branching to promote iTreg over T_H_17 differentiation (Figure 2)" is not directly supported by the data. Without IL-2, Tregs die, with or without highly branched glycans.*

Thank you for this point. Figure 2 only showed a ratio of T_H_17 to iTreg, where the results could have been explained by only a change in Treg death from blockade of IL-2. We have now separated the data to demonstrate that anti-IL-2 antibody enhanced production of T_H_17 cells while also reducing the number of iTreg’s despite treatment with GlcNAc (Figure 2—figure supplement 1). Moreover, we showed that addition of exogenous IL-2 enhanced iTreg and inhibited T_H_17 in the presence but not absence of branching (Figure 2, Figure 2—figure supplement 1), consistent with branching regulating a change in cell fate rather than simply death of iTreg via IL-2 signaling.

The authors conclude that branching drives CD25 surface retention, but they only measure CD25 surface concentration. There is no direct data showing anything about surface retention. It is possible that the expression is merely lower. It is possible that the Mgat1 mutation creates problems with protein folding (a common effect). It is possible that it never even reaches the cell surface.

Using an “acid wash” endocytosis assay that we have used previously, we have added new data indicating that CD25 surface retention is reduced by loss of branching (Figure 2—figure supplement 1).

*For readers not familiar with the Warburg effect and aerobic glycolysis, it would be helpful to mention that glycolysis is the conditional norm of anaerobic metabolism, and that the discovery by Warburg was that cancer cells constitutively use glycolysis even in aerobic conditions (e.g. aerobic glycolysis).*

We have added this point to the Introduction.